# Changes in the Epidemiology of Hepatocellular Carcinoma in Carinthia, Austria, 2012–2023

**DOI:** 10.3390/cancers15215215

**Published:** 2023-10-30

**Authors:** Florian Hucke, Heleen Emmer, Roberto Emmer, Miriam Hucke, Simona Bota, Matthias Fürstner, Klaus Hausegger, Reinhard Mittermair, Markus Peck-Radosavljevic

**Affiliations:** 1Department of Internal Medicine and Gastroenterology (IMuG), Hepatology, Endocrinology, Rheumatology and Nephrology Including Emergency Medicine (ZAE), Klinikum Klagenfurt am Wörthersee, 9020 Klagenfurt, Austria; heleen.emmer@outlook.com (H.E.); roberto.emmer@kabeg.at (R.E.); miriam.hucke@kabeg.at (M.H.); simona.bota@kabeg.at (S.B.); markus.peck-radosavljevic@kabeg.at (M.P.-R.); 2Institute for Diagnostic and Interventional Radiology, Klinikum Klagenfurt am Wörthersee, 9020 Klagenfurt, Austria; matthias.fuerstner@kabeg.at (M.F.); klaus.hausegger@kabeg.at (K.H.); 3Department of General and Abdominal Surgery, Klinikum Klagenfurt am Wörthersee, 9020 Klagenfurt, Austria; reinhard.mittermair@kabeg.at

**Keywords:** hepatocellular carcinoma, survival HCC, epidemiology, NASH, Austria

## Abstract

**Simple Summary:**

Hepatocellular carcinoma (HCC) is a type of primary liver malignancy. Worldwide, HCC is one of the leading causes of cancer-related deaths in adults. Liver cancer is currently the sixth most common cancer worldwide. Many changes in the epidemiology and underlying etiology of HCC have taken place in recent years. In industrialized countries, NAFLD/NASH has been playing an increasingly important role in the development of HCC. Despite major advances in the systemic medical management and availability of immunotherapy, no marked improvement in overall survival time has been found over the past decade.

**Abstract:**

Background: Hepatocellular carcinoma (HCC) is one of the leading causes of cancer-related deaths and remains a major burden on healthcare systems worldwide. The incidence of HCC continues to rise globally, despite preventative efforts being made. Aims: This study aimed to investigate epidemiological changes observed in the etiology and survival outcomes of HCC patients at Klinikum Klagenfurt am Wörthersee between 2012 and 2023. Methods: This was a retrospective, single-center cohort study. Two time-periods (2012–2017 and 2018–2023) were created to enable comparison between the respective intervals. IBM SPSS was used to analyze statistical data. Results: More patients were diagnosed with HCC during the second time period (*n* = 128, *n* = 148). The median age of diagnosis was 72.5 years (SD 8.6). Patients were on average 2 years younger in the second time period compared to the first (*p* = 0.042). Alcohol remained the leading underlying etiology of HCC and no statistically significant change was seen over time (*p* = 0.353). Nevertheless, a clear upward trend in the number of NASH cases was evident over time (*n* = 15, *n* = 28, respectively). Nearly half of the patient population had a normal AFP (<7 µg/L) level at the time of diagnosis (*n* = 116, 42.6%). The survival time for HCC patients remained similar between time periods, with a median overall survival time of 20.5 months (95% CI 16.8–24.2, *p* = 0.841), despite improvements in management strategies and the availability of new systemic treatments. More advanced-stage HCC cases were documented in the second period (BCLC-C, *n* = 23 to *n* = 46, *p* = 0.051). An increased number of HCC patients without liver cirrhosis were identified during the second time period (*n* = 22, *n*= 47, respectively, *p* = 0.005). NASH was the most common underlying etiology in patients without liver cirrhosis (50%) compared to alcohol use in being the primary cause in cirrhotic patients (65%, *p* < 0.001). Conclusion: HCC continues to be an important health concern in our society. The number of HCC patients without liver cirrhosis is steadily increasing, with NAFLD/NASH, due to underlying lifestyle diseases playing an important etiological role. Continued efforts should be made to prevent HCC and to screen at-risk population groups. Preventative strategies and screening techniques should be adjusted in light of the changing epidemiological landscape of HCC, where more focus will have to be placed on detecting HCC in patients without underlying cirrhosis.

## 1. Introduction

### 1.1. Background and Epidemiology

Hepatocellular carcinoma (HCC) is a type of primary liver malignancy. The majority (90%) of primary liver cancer cases are attributed to HCC [1,2]. Worldwide, HCC is one of the leading causes of cancer-related deaths in adults. According to the World Health Organization’s (WHO) Global Cancer Observatory (GLOBOCAN), a total of 905,677 new cases of liver cancer and 830,180 deaths due to liver cancer were recorded in 2020. This accounts for 4.7% of all new cancer cases in 2020 and 8.3% of all cancer-related deaths [3]. Liver cancer is currently the sixth most common cancer worldwide. It is predicted that the number of new cases of primary liver cancer will rise drastically between 2020 and 2040. 1.4 million new diagnoses are anticipated to occur in 2040 [4]. 

### 1.2. Etiology and Risk Factors

HCC mainly develops in the setting of underlying liver cirrhosis [1]. Hepatitis B virus (HBV) is still the leading underlying cause of HCC globally, due to its high prevalence in Asia and Africa [1]. Hepatitis C virus (HCV) is another important virological cause of liver cirrhosis, especially in Northern Africa and Asia [5]. In recent years, the use of direct-acting antiviral (DAA) therapy has shown very promising results in reducing the risk of HCC development. In patients with a sustained virological response (undetectable HCV RNA, 6 months after completing treatment), the risk of HCC development reduces by 50–80% [6]. In the West, nonalcoholic fatty liver disease (NAFLD) and nonalcoholic steatohepatitis (NASH), associated with metabolic lifestyle diseases, are rapidly becoming major role-players in the etiological development of HCC [2]. HCC may develop without the presence of liver cirrhosis in an estimated 20–30% of NASH cases [7]. Chronic alcohol use disorder is another well-known risk factor for liver cirrhosis. The estimated risk of developing HCC in patients with decompensated cirrhosis due to alcohol use is around 1% per year [8].

### 1.3. HCC in Austria

Between 2010 and 2018, a total of 7146 individuals were diagnosed with HCC in Austria, of which 75% were male and 25% were female [9]. A total number of 1114 new liver cancer cases were documented in the 2020 GLOBOCAN database, with 993 deaths reported. Liver cancer ranked as the 14th most commonly diagnosed cancer and obtained the 6th place for highest number of deaths caused by cancer in Austria [10]. 

In Austria, alcohol remains the predominant cause of liver cirrhosis. According to the textbook of alcohol in Austria, the number of hospital admissions for alcohol dependency have however steadily declined over the past decade [11].

### 1.4. Staging Systems

The Barcelona Clinic Liver Cancer (BCLC) system is the most frequently used staging tool in clinical practice today. The European Association for the Study of the Liver (EASL) recommends the updated BCLC staging system for the prediction of prognosis and the allocation of appropriate treatment [12]. In the 2022 version of the BCLC model, liver function is determined more thoroughly and not just through the conventional Child–Pugh classification. The albumin–bilirubin score and AFP concentration are examples of the additions made to better determine compensated liver function.

Furthermore, newly established therapeutic options in systemic therapies have been included. Between 2018 and 2019, the IMbrave150 trail compared Sorafenib to the Atezolizumab/Bevacizumab combination and proved a better overall survival rate with the latter treatment strategy [13].

### 1.5. Study Objectives

In this study, we aimed to investigate the changes observed in the epidemiology and etiology of HCC at Klinikum Klagenfurt between 2012 and 2023.

A rise in NAFLD/NASH leading to HCC has been observed in other developed countries with limited data available on the incidence of NASH within our more rural region [2]. We therefore aimed to examine the changes in underlying etiology of HCC to determine whether this is also true in our setting. If a change in the incidence of etiological factors has occurred, it could mean that a gap in current screening programs may exist. 

We furthermore aimed to investigate the survival outcome and prognostic parameters in rural HCC patients. We also aimed to investigate whether or not overall survival has improved over time, after immunotherapeutic agents and more TKI therapies became widely available in 2018.

Given that the incidence of HCC is rising worldwide, with HCC having a poor prognosis when diagnosed at a late stage, epidemiological data should be used to improve prevention strategies by providing the necessary information in which patient surveillance is lacking.

## 2. Materials and Methods

### 2.1. Single-Centre Cohort Study

This was a single-center cohort study at Klinikum Klagenfurt am Wörthersee in Austria. Klinikum Klagenfurt acts as a tertiary hospital for the province of Carinthia and serves a large rural drainage area. Data were obtained retrospectively from the hospital’s clinical information software system and the laboratory results were acquired from the hospital’s electronic system. Each patient’s electronic file was reviewed individually in order to obtain the desired datapoints. Data were collected in an Excel database and kept anonymous by allocating numbers to patient names.

Several patient characteristics (age, sex, BMI, time of HCC diagnosis, comorbidities, date of last follow-up or death), clinical features (ascites, ECOG performance status, presence of hepatic encephalopathy), tumor characteristics (size and number of nodules, focality, macrovascular invasion, distant or lymph-node metastases, histological grading), laboratory values (alpha-fetoprotein in µg/L, albumin in g/dL, bilirubin in mg/dL, INR and prothrombin time, C-reactive protein in mg/dl, thrombocytes in G/L aspartate transaminase in U/L, alanine transaminase in U/L), and therapeutic treatments for HCC were documented.

Two time periods were established to evaluate the changes in epidemiology and survival. The first time period included all patients diagnosed with HCC from January 2012 until the end of 2017 and the second comprised patients diagnosed between the beginning of 2018 until February 2023. In 2018, a marked change in the systemic medical management of HCC started to take place and 2018 was therefore used as the cutoff for comparison. By creating two time periods, we were able to compare changes in etiology and survival outcomes that had occurred over time. We therefore needed to create two time periods in order to comment on whether or not change had occurred. Regarding survival outcome, we first calculated the survival time for the cohort as a whole to provide a general overview; thereafter, we analyzed the changes that occurred over time. 

Liver cirrhosis was diagnosed by means of radiology (ultrasound, CT, MRI), laboratory parameters according to Baveno, Fibroscan, or histologically. First-line treatment was divided into five categories: surgical, which consisted of orthotopic liver transplantation (OLT) and partial hepatectomies; interventional therapy, which included ablation techniques with radiofrequency ablation (RFA) or microwave ablation (MWA); and transarterial chemoembolization (TACE). The fourth category consisted of patients who had received systemic therapy (tyrosine kinase inhibitors or other immunotherapies) as a first-line treatment, and the fifth category included patients who had received best supportive care as first-line treatment. 

All patients above the age of 18 diagnosed with HCC either histologically by means of a liver biopsy or through multiphase imaging techniques (quadruple-phase CT or contrast-enhanced MRI) were captured in our data collection. Patients with mixed HCC and biliary tract cancers (BTCs) and patients with missing data vital to our study aim were excluded from our study population. 

### 2.2. Statistics

The IBM SPSS version 25.0 (IBM Corp., Armonk, NY, USA) was used to analyze the captured data from the selected patient population. Nominal data were shown as absolute numbers (*n*) and percentages (%) and numerical data were presented as the median with the standard deviation range included. Pearson’s chi-square test was applied to analyze categorical data and to determine how likely it was for differences between datasets to occur due to chance. A *p*-value of <0.050 was seen as statistically significant. Cox regression analysis and the Kaplan–Meier method were used to estimate and analyze survival probability. 

This retrospective data analysis was approved by the local ethics committee (EK A30/18).

## 3. Results

From January 2012 to February 2023, a total of 285 patients were diagnosed with HCC. A total of 276 patients met the requirements of the inclusion criteria and were considered for further statistical analysis.

### 3.1. Descriptive Patient Analysis and Time Period Comparison (Table 1)

The first time period consisted of 128 patients (46.4% of the total patient population) and the second of 148 patients (53.6%). The incidence rate was 5.9 times higher in males compared to females. In the first time period, the median age of diagnosis was 73 years (SD 8.6) compared to 71 years (SD 8.5) in the second time period (*p* = 0.042).

During the first time period, 22 patients (17.2%) had HCC without underlying liver cirrhosis. This number increased to 47 (31.8%) during the second time period (*p* = 0.005). 

Alcoholic liver disease was the most common underlying etiology, with a total of 139 patients (55.6%). Fifty-five patients (22%) had viral hepatitis and forty-three (17.2%) had underlying NASH. A total of 13 patients (5.2%) were classified as “other” (e.g., hemochromatosis, Budd–Chiari syndrome, and primary biliary cholangitis). There was no statistically significant change observed in the etiology of HCC between the two time categories (*p* = 0.353).

Alpha-fetoprotein (AFP) values were documented for 272 patients. A total of 116 patients (42.6%) had unelevated AFP levels (<7 ng/mL) and 156 (57.4%) had a raised AFP (≥7 ng/mL) level at the time of diagnosis. No difference could be observed between the various etiologies. Only 66 patients (24.3%) had an elevated AFP level ≥200 ng/mL, and 56 patients (20.6%) had an AFP level of ≥400 ng/mL. No significant change in AFP levels was observed over time (*p* = 0.199), despite the shift in etiology of the underlying liver disease.

Concerning the Child–Pugh scoring system, a total of 116 patients (42%) were classified as Child–Pugh A. 

With regard to tumor staging, the majority of patients (40.6%, *n* = 112) were classified as BCLC-A at the time of diagnosis. The number of patients diagnosed at the advanced stage BCLC-C increased over time (*p* = 0.051).

Regarding the ALBI score, 30.6% (*n* = 83) of HCC patients were staged as ALBI grade 1. Significantly more patients were staged as ALBI grade 1 in the second time period (*p* < 0.001, Table 1). 

There was no noteworthy change in histological grading over time (*p* = 0.196). The median size of the largest space-occupying lesion was 4.5 cm in both time periods. No statistically significant difference in the incidence of portal vein thrombosis, macrovascular invasion, enlarged lymph nodes, or distant metastases was observed over time. 

Concerning comorbidities, no significant change was seen over time. The mean body mass index (BMI) for all patients was 28 (SD 4.5). The mean BMI of HCC patients at the time of diagnosis did not change over time (*p* = 0.871).

In our patient population, a total of seven orthotopic liver transplantations (OLTs) were performed in the first time period and six in the second (*p* = 0.580). 

Regarding first-line therapy for all patients, 15% (*n* = 42) of patients underwent surgical treatment (OLT/resection), 16% (*n* = 45) underwent ablation by means of RFA/MWA, 28% (*n* = 76) underwent TACE, 27% (*n* = 75) obtained systemic medical therapy (TKI/IT), and 14% (*n* = 38) of patients received best supportive care.

While surgical treatment remained stable over time, there was a significant shift from interventional to systemic treatments between the first and second observation period (*p* < 0.001). 

**Table 1 cancers-15-05215-t001:** Descriptive parameters comparing the two time periods.

		Time Period 1 2012–2017 (*N* = 128)	Time Period 2 2018–2023 (*n* = 148)	
Variable		N (%)	N (%)	*p*-Value
Age	Mean ± STD	73 ± 8.6	71 ± 8.5	0.042
Sex	Male	109 (85)	127 (86)	
	Female	19 (15)	21 (14)	0.878
BMI	Mean ± STD	28 ± 4.5	28 ± 5.3	0.871
Liver cirrhosis	Present	106 (83)	101 (68)	
	Absent	22 (17)	47 (32)	0.005
Etiology	Alcohol	69 (59)	70 (53)	
	Viral	28 (24)	27 (21)	
	NASH	15 (13)	28 (21)	
	Other	6 (5)	7 (5)	0.353
Ascites	Present	28 (22)	41 (28)	
	Absent	98 (78)	106 (72)	0.283
Child–Pugh	No cirrhosis	22 (17)	47 (32)	
	A	55 (43)	61 (41)	
	B	38 (30)	27 (18)	
	C	13 (10)	13 (9)	0.020
BCLC stage	A	53 (41)	59 (40)	
	B	32 (25)	29 (20)	
	C	23 (18)	46 (31)	
	D	20 (16)	14 (10)	0.051
ALBI score	Grade 1	18 (15)	65 (44)	
	Grade 2	81 (66)	67 (45)	
	Grade 3	24 (20)	16 (11)	< 0.001
Grading	1	30 (43)	40 (37)	
(Edmondson–Steiner)	2	28 (40)	53 (50)	
	3	11 (16)	14 (13)	
	4	1 (1)	0 (0)	0.196
	Initially negative	10	6	
Focality	Unifocal	55 (43)	72 (49)	
	Multifocal	73 (57)	76 (51)	0.345
SOL (size in cm)	Mean ± STD	5.7 ± 3.9	5.4 (3.5)	0.433
Up-to-seven	≤7	62 (49)	72 (49)	
	>7	65 (51)	76 (51)	0.978
Macrovascular invasion	Present	22 (17)	36 (24)	
	Absent	105 (83)	112 (76)	0.156
Enlarged lymph nodes (>2 cm)	Present	23 (18)	18 (12)	
	Absent	103 (82)	127 (88)	0.181
Distant metastases	Present	9 (7)	18 (12)	
	Absent	118 (93)	130 (88)	0.158
Coronary artery disease	Present	13 (10)	21 (14)	
	Absent	112 (90)	127 (86)	0.345
Hypertension	Present	80 (64)	87 (59)	
	Absent	45 (36)	61 (41)	0.378
Diabetes	Present	38 (30)	50 (34)	
	Absent	90 (70)	98 (66)	0.466
AFP	Raised (>7 µg/L)	76 (61)	80 (54)	
	Normal	49 (39)	67 (46)	0.289
First-line therapy	Surgical/OLT	21 (16)	21 (14)	
	MWA/RFA	13 (10)	32 (22)	
	TACE	55 (43)	21 (14)	
	Systemic (IT/TKI)	16 (13)	59 (40)	
	BSC	23(18))	15/19	<0.001

Abbreviations: STD: standard deviation; BMI: body mass index; SOL: space-occupying lesion; AFP: alpha-fetoprotein; OLT: orthotopic liver transplantation; RFA: radiofrequency ablation; MWA: microwave ablation; TACE: transarterial chemoembolization.

In a more detailed look at BCLC-B patients, a treatment migration away from surgical/interventional treatment towards more systemic therapy was evident (four patients (12.5%) in the first time period compared to two patients (6.9%) in the second time period). This occurrence cannot be evaluated statistically due to the low number of patients, as surgery is not a standard form of treatment for BCLC-B patients. However, 26 (81.3%) patients received ablative/interventional therapy in the first time period, compared to 11 (37.9%) patients in the second (*p* < 0.001), and only 2 (6.3%) patients received systemic drug treatment in the first time period, compared to 16 (55.2%) patients in the second (*p* < 0.001, Table 1).

### 3.2. Univariate Analysis of Median Survival Times (Table 2 and Appendix A)

The overall survival time was first calculated for the collective cohort to provide an overview of survival outcome (see Table 3). We thereafter also statistically analyzed the change in survival time between the two time periods in order to assess whether survival had improved after the introduction of new immunotherapeutic agents (see Appendix A).

The median overall survival was 20.5 months, with 21.9 months (95% CI 17–26.8) in the first time period and 20.2 months (95% CI 15.7–24.6) in the second time period (*p* = 0.841). 

When patients who had undergone OLTs were removed from the analysis, the median overall survival was 19.4 months (95% CI 15.9–22.9), with no significant change between the two time intervals (*p* = 0.902).

The median OS for HCC patients without liver cirrhosis was 22 months (95% CI 8.7–35.7) and those with cirrhosis had an OS of 20.2 months (95% CI 16.9–23.4, *p* = 0.262). No significant change in survival time for cirrhotic vs. noncirrhotic patients were found between time periods (*p* = 0.279). 

Regarding survival and etiology, patients with alcohol as underlying cause had the shortest median OS of 18.1 months (95% CI 14.2–22). Those with viral hepatitis had a median OS of 32.2 months (95% CI 15.9–48.3). Patients with NASH had a median OS of 38.7 months (95% CI 18.4–59) and those with other causes had a median OS of 19.7 months (95% CI 0–53.7, *p* = 0.004). No difference in OS for alcohol (*p* = 0.772), viral (*p* = 0.868), or NASH (*p* = 0.970) as underlying etiology was found in cirrhotic compared to noncirrhotic patients.

Patients with BCLC-A had a median OS of 34.9 months (95% CI 15.8–53.9), those with BCLC-B had 27 months (95% CI 17.9–36.2), BCLC-C had 10 months (95% CI 7.3–12.8), and BCLC-D a median OS of 1.9 months (95% CI 1.3–2.6) (*p* < 0.001) (see Figure 1B and Figure 2B).

Looking at patients with early-stage (BCLC-A) HCC and advanced-stage (BCLC-C) separately, OS in the first period for BCLC-A was 33.1 months (95% CI 11.2–55.1) and in the second period 53 months (95% CI 20.1–85.5; *p* = 0.805). The OS for BCLC-C in the first time period was 10.4 months (95% CI 5.9–15) and 10 months in the second time period (95% CI 6.2–14, *p* = 0.446).

No significant difference in overall survival for patients with cirrhosis versus non-cirrhosis in BCLC-A and BCLC-C, respectively, was found (*p* = 0.977; *p* = 0.449).

According to the Child–Pugh classification, the median OS for patients staged as Child–Pugh A was 27 months (95% CI 19.4–34.6), for Child–Pugh stage B 14.6 months (95% CI 10.1–19), and 2.6 months (95% CI 0.2–5) for Child–Pugh C (*p* < 0.001) (see Figure 1A and Figure 2A).

Patients classified as Child–Pugh A had a survival time of 28.7 months (95% CI 19.8–37.6) in the first time period compared to 27 months (95% CI 12.8–41.3) in the second time period. Patients with Child–Pugh stage B had a survival time of 15.2 months (95% CI 9.3–21) and 11.8 months (95% CI 0.9–22.7) in the respective time periods and Child–Pugh C a survival time of 1.8 months (95% CI 0–3.6) in the first and 5.6 months (95% CI 0–13.7) in the second time period (*p* ≤ 0.001).

Concerning the ALBI score, patients graded as ALBI 1 had a median survival time of 34.6 months (95% CI 26–34.1). For patients classified as ALBI grade 2, the median survival time was 19.4 months (95% CI 15.1–23.6) and 6.6 months (95% CI 1–12.2) for ALBI grade 3. There is a significant change over time for survival of the various ALBI grades (*p* < 0.001) (see Figure 1C and Figure 2C).

Patients with a normal AFP level (<7) had a median survival time of 34.6 months (95% CI 22.8–46.3) and those with a raised AFP above 7 had a 14.6-month median survival time (95% CI 10.3–18.9). Patients with an AFP serum level >7 and ≤200 ng/mL had a median survival time of 20.7 months (95% CI 16.1–25.4). Those with an AFP >200 and ≤400 had a median survival time of 5.2 months (95% CI 3.2–7.2) and those with an AFP ≥ 400 had a 7.3 month (95% CI 4.4–10.3) survival time (*p* < 0.001). 

The median survival time for those who fulfilled the up-to-seven criteria (≤7 points) was 33.1 months (95% CI 17.9–48.4), while those with >7 points had a median survival time of 11.8 months (95% 9.6–14.1, *p* < 0.001). 

There was no significant difference in OS observed in the different comorbidities or concomitant medication.

**Table 2 cancers-15-05215-t002:** Descriptive statistics and univariate analysis of patients diagnosed at Klinikum Klagenfurt between 2012 and 2023.

Variable		*N* = 276	Overall Survival (Months)		*p*-Value
			Median	95% CI	(Log Rank)
Age	<70	104	18.7	12.7–24.7	
	≥70	172	22.2	16.8–27.6	0.163
Sex	male	236	21.6	17.9–25.3	
	female	40	16.5	9.5–23.4	0.842
Liver cirrhosis	Present	207	20.2	1.6–16.9	0.262
	Absent	69	22	8.4–35.7	
Etiology	Alcohol	139	18	14.2–22	
	Viral	55	32.2	15.9–48.4	
	NASH	43	38.7	18.4–59	
	Other	13	19.7	0–53.7	0.004
Ascites	Present	69	7.4	1.5–13.3	
	Absent	204	26.2	21.5–31	<0.001
Child–Pugh	No cirrhosis	69	22	8.4–35.7	
	A	116	27	19.4–34.6	
	B	65	14.6	10.1–19	<0.001
	C	26	2.6	0.2–5	
BCLC	A	112	34.8	15.8–53.9	
	B	61	27	17.9–36.2	
	C	69	10	7.3–12.8	<0.001
	D	34	1.9	1.3–2.6	
Focality	Unifocal	127	27	19–35.1	
	Multifocal	149	17.1	11.6–22.5	0.002
Up-to-seven	≤7	141	33.1	17.9–48.4	
	>7	134	11.8	9.6–14.1	<0.001
SBL	Present	23	9.1	3.8–14.4	
	Absent	252	22	17.9–26.5	<0.001
Macrovascular invasion	Present	58	5.3	3.2–7.6	
	Absent	217	26.9	21–32.8	<0.001
Lymph node enlargement (>2 cm)	Present	41	6.8	1.9–11.8	
	Absent	230	24.2	18.9–29-5	<0.001
CRP	Raised	120	11.8	8.6–15.1	
	Normal	152	32.2	23.4–40.9	<0.001
Coronary artery disease	Present	34	19.4	2.1–36.6	
	Absent	239	20.7	16.8–24.7	0.463
Chronic kidney disease	Present	29	15.6	8.7–22.5	
	Absent	243	20.7	17–24.5	0.310
Hypertension	Present	167	20.7	16–25.5	
	Absent	106	19.7	15–24.6	0.880
Diabetes mellitus	Present	88	20.2	16.2–24.1	
	Absent	188	21.2	15–27.5	0.605
BMI	<25	75	19.8	13.6–26	
	≥25	168	24.2	18.7–29.6	0.676
AFP	Raised (>7)	156	14.6	10.3–18.9	
	Normal	116	34.6	22.8–46.3	<0.001
ALBI score	Grade 1	83	34.6	26–43.1	
	Grade 2	148	19.2	15.1–23.6	
	Grade 3	40	6.6	1–12.2	<0.001
First-line therapy	Surgical	31	63.2	40.9–85.5	
	Interventional	132	28.7	22–35.4	
	Systemic	72	11.8	9.7–14	
	None	40	2	1.1–3.1	<0.001
ASA	Present	59	26.9	14.6–39.1	
	Absent	217	19.8	16.2–23.4	0.264

Abbreviations: BCLC: Barcelona Clinic Liver Cancer Staging; CRP: C-reactive protein; BMI: body mass index; AFP: alpha-fetoprotein; ALBI: albumin–bilirubin score; SBL: sclerotic bone lesions; ASA: acetylsalicylic acid.

As survival time varies by etiology of HCC, we further looked at the different treatment categories for each etiology (see Table 3 for results). Except for MWA and RFA, etiology had no prognostic impact on overall survival according to the different treatments.

**Table 3 cancers-15-05215-t003:** Survival for treatment by etiology.

Variable		Total *N*	Overall Survival		*p*-Value
			Median	95% CI	(Log Rank)
Resection/OLT	Alcohol	15	56.4	0.0–113.3	
	Viral	8	70.1		
	Other	14	46.0	31.1–60.8	0.825
MWA/RFA	Alcohol	18	20.2	9.7–30.6	
	Viral	14	53.3	8.2–98.4	
	Other	12			0.003
TACE	Alcohol	46	24.0	21.1–26.9	
	Viral	12	27.0	22.1–32.0	
	Other	13	24.2	15.5–32.9	0.443
Systemic (TKI/IT)	Alcohol	36	11.8	9.1–14.6	
	Viral	14	16.2	2.7–29.8	
	Other	13	12.8	8.6–17.0	0.477
BSC	Alcohol	24	1.9	0.8–2.9	
	Viral	7	1.8	0.1–3.5	
	Other	4	6.8	0.5–13.0	0.330

Abbreviations: OLT: orthotopic liver transplant; MWA: microwave ablation; RFA: radiofrequency ablation; TACE: transarterial chemoembolization; TKI: tyrosine kinase inhibitor; IT: immunotherapy; BSC: best supportive care.

### 3.3. Statistical Analysis of HCC Patients without Liver Cirrhosis (Table 4)

The number of patients diagnosed with HCC where no liver cirrhosis was present increased between the first time interval and the second. A more in-depth look at the specific characteristics of HCC patients without underlying liver cirrhosis was taken to better understand why this phenomenon occurred.

The male-to-female ratio and median age were similar in the noncirrhotic cohort compared to the cirrhotic cohort. 

With regards to etiology in noncirrhotic patients, 7 patients (15.2%) were known with alcoholic liver disease, 14 patients (30.4%) had underlying viral hepatitis, 23 (50%) had NASH, and 2 patients (4.3%) were classified as other. In patients with liver cirrhosis, 64.7% (*n* = 132) of patients had underlying alcoholic liver disease, 20.1% (*n* = 41) had viral hepatitis, 9.8% (*n* = 20) had NASH, and 5.4% (*n* = 11) of patients were classified as other (*p* < 0.001).

The largest percentage of noncirrhotic patients were staged as BCLC-A (*n* = 29, 42%), similar to the 40% (*n* = 83) of cirrhotic patients who were also diagnosed as BCLC-A. 21.7% (*n* = 15) of noncirrhotic patients were classified as BCLC-B, as was the case with 22% (*n* = 46) of cirrhotic patients. Of the total, 34.8% (*n* = 24) was staged as BCLC-C and 1.4% (*n* = 1) as BCLC-D, which was in clear contrast to cirrhotic patients, where 22% (*n* = 45) was BCLC stage V and 16% (*n* = 33) was BCLC-D (*p* = 0.006).

Only 47.1% (*n* = 32) of patients without cirrhosis had a raised AFP (≥7) compared to 60.8% (*n* = 124) of patients with cirrhosis who had a raised AFP level (*p* = 0.047).

The fibrosis-4 score was calculated for all noncirrhotic patients to estimate the severity of fibrosis. Out of the 69 patients, 6 (8.7%) had mild fibrosis, 24 (34.8%) had moderate fibrosis, and 39 (56.5%) had severe fibrosis. 

The median size of the largest space-occupying lesion was 5.3 cm (SD 3.4) in noncirrhotic patients, compared to 4 cm (SD 3.7) in cirrhotic patients (*p* = 0.149).

Eleven patients (15.9%) had PVT at the time of diagnosis, nine patients (13%) had distant/extrahepatic metastases, thirteen patients (18.8%) had MVIs, thirteen patients (18.8%) had enlarged lymph nodes (>2 cm), and nine (13%) had SBLs. Patients without liver cirrhosis predominantly (68.1%) had lesions restricted to one lobe of the liver.

Apart from arterial hypertension, there was no difference in comorbidities between patient with and without cirrhosis. 

Surgical treatments and systemic therapies including immunotherapy were significantly more often performed in noncirrhotic patients, while interventional treatments were more common in cirrhotic patients (*p* > 0.001).

**Table 4 cancers-15-05215-t004:** HCC patients with liver cirrhosis compared to noncirrhotic patients.

Variable		Cirrhosis (*N* = 207)	Non-Cirrhosis (*n* = 69)	
		*N* (%)	*N* (%)	*p*-Value
Age	Mean ± STD	71 ± 8.6	73 ± 8.8	0.112
Sex	Male	178 (86)	58 (84)	
	Female	29 (14)	11 (16)	0.693
BMI	Mean ± STD	28 ± 5.1	28 ± 4.8	0.699
Etiology	Alcohol	132 (65)	7 (15)	
	Viral	41 (20)	14 (30)	
	NASH	20 (10)	23 (50)	
	Other	11 (5)	2 (4)	<0.001
Ascites	Present	68 (33)	1 (1)	
	Absent	136 (67)	68 (99)	<0.001
BCLC stage	A	83 (40)	29 (42)	
	B	46 (22)	15 (22)	
	C	45 (22)	24 (35)	
	D	33 (16)	1 (1)	0.006
ALBI score	Grade 1	41 (20)	42 (62)	
	Grade 2	124 (61)	24 (35)	
	Grade 3	38 (19)	2 (3)	<0.001
Grading	0/negative	16 (12.5)	0 (0)	
	1	39 (30.5)	31 (48)	
	2	55 (43)	26 (40)	
	3	17 (13)	8 (12)	
	4	1 (1)	0 (0)	0.015
Focality	Unifocal	88 (42.5)	39 (56.5)	
	Multifocal	119 (57.5)	30 (43.5)	0.043
SOL (size in cm)	Mean ± STD	5.4 ± 3.7	6.1 ± 3.4	0.149
Up-to-seven	≤7	110 (53)	31 (45)	
	>7	96 (47)	38 (55)	0.223
Macrovascular invasions	Present	45 (22)	13 (19)	
	Absent	161 (78)	56 (81)	0.597
Enlarged lymph nodes (>2 cm)	Present	28 (14)	13 (19)	
	Absent	174 (86)	56 (81)	0.319
Coronary artery disease	Present	24 (12)	10 (14.5)	
	Absent	180 (88)	59 (85.5)	0.553
Hypertension	Present	116 (57)	51 (74)	
	Absent	88 (43)	18 (26)	0.012
Diabetes	Present	67 (32)	21 (30)	
	Absent	140 (68)	48 (67)	0.765
AFP	Raised (>7)	124 (61)	32 (47)	
	Normal	80 (39)	36 (53)	0.047
First-line therapy	Surgical	15 (7)	16 (23)	
	Interventional	109 (53)	23 (33)	
	Systemic	45 (22)	27 (39)	
	None	37 (18)	3 (4)	<0.001
OLT	Yes	12 (6)	1 (1)	
	No	195 (94)	68 (99)	0.140
Resection	Yes	13 (6)	14 (20)	
	No	194 (94)	55 (80)	0.001
RFA (first-line)	Yes	49 (24)	10 (14.5)	
	No	158 (76)	59 (85.5)	0.107
MWA (first-line)	Yes	14 (7)	3 (4)	
	No	193 (93)	66 (96)	0.470
TACE (first-line)	Yes	86 (41.5)	16 (23)	
	No	121 (58.5)	53 (77)	0.006
Immunotherapy	Yes	31 (15)	21 (30)	
	No	176 (85)	48 (67)	0.004

Abbreviations: HCC: hepatocellular carcinoma; STD: standard deviation; BMI: body mass index; SOL: space-occupying lesion; AFP: alpha-fetoprotein; OLT: orthotopic liver transplantation; RFA: radiofrequency ablation; MWA: microwave ablation; TACE: transarterial chemoembolization.

## 4. Discussion

The incidence of HCC is known to be rising globally and the same was found in our cohort over time [14,15]. Although underlying liver cirrhosis is known to be an important role player in the development of HCC, it is interesting to note that the proportion of HCC patients without liver cirrhosis notably increased in the second time period [1].

In contrast to HCC patients with liver cirrhosis, our data provided evidence for a significant difference in the underlying cause of HCC in noncirrhotic patients. Whereas HCC in existing cirrhosis was predominately found in the setting of alcoholic liver disease (known to be the most prevalent underlying cause in Austria), NASH was the cause of HCC in half of noncirrhotic patients and a clear upward trend in the number of NASH patients was demonstrated over time [11]. In a large metanalysis performed by Stine et al., it was shown that in noncirrhotic patients, those with NASH had a higher risk of developing HCC [16].

The nonalcoholic fatty liver disease process starts with a nonalcoholic fatty liver (NAFL), which may worsen and develop into nonalcoholic steatohepatitis (NASH). NASH is characterized by steatosis and accompanied by hepatocyte inflammation, which may include fibrosis [17]. Most noncirrhotic patients in our cohort had severe fibrosis according to the Fib-4 score. Patients with NAFLD are often found to also suffer from other components of the metabolic syndrome. One proposed hypothesis for the development of HCC in noncirrhotic NASH patients is the development of malignant HCC from a hepatocellular adenoma in the setting of metabolic syndrome [18,19].

Despite the increase in HCC in noncirrhotic patients, comorbid lifestyle diseases generally known as predisposing factors for the development of NASH did not significantly increase over time. It is, however, important to note that we still had a much higher incidence rate of diabetes in our cohort compared to the general population. Currently, the global incidence of diabetes in adults is around 10.5%, compared to 32% of patients in our cohort [20]. The same is true for hypertension, as the worldwide incidence is around 30% of the population, compared to 61% of patients in our cohort [21]. The incidence of hypertension was also significantly higher in the noncirrhotic group compared to cirrhotic. Although most patients in the overall cohort were overweight (BMI ≥ 25) at the time of diagnosis, no significant difference was evident between the cirrhotic and noncirrhotic group. Given the significantly higher rate of ascites in the cirrhotic patient group, it may have led to an overestimation of the true “dry” weight and therefore masked a higher BMI in noncirrhotic patients.

With regard to OS, patients in our cohort survived between 20 to 22 months. We did not observe any further difference in survival between cirrhotic and noncirrhotic patients. The similar OS rate of cirrhotic vs. noncirrhotic patients can be explained by the more advanced BCLC stage of noncirrhotic patients at the time of diagnosis. This finding highlights the lack of surveillance protocols for noncirrhotic patients.

AFP remains a common screening parameter used in practice today, but nearly half (43%) of patients did not have a raised AFP (<7 ng/mL) at the time of diagnosis. Furthermore, AFP was only elevated in 47% of the noncirrhotic patients, compared to 60% of cirrhotic patients. Our data are corroborated by a review written by Desai et al., where it was also found that AFP levels were only elevated in 31–67% of noncirrhotic cases, compared to 63–84% in cirrhotic patients [18]. This finding might be attributed to the increase in serum AFP levels by liver cirrhosis itself. The combined use of ultrasound and AFP levels improves the sensitivity of HCC detection and is currently the screening regime of choice [22]. Ultrasound screening should be performed biannually, and further imaging is warranted if either a lesion of ≥1 cm is detected or a lesion of <1 cm is accompanied by a rising AFP value [23]. AFP levels, however, still correlated with OS and add a valuable contribution in predicting prognosis, as proven in numerous other studies [24,25,26].

To improve the early detection of HCC and surveillance of cirrhotic patients, Klinikum Klagenfurt introduced educational programs in the region after becoming an expert primary center for HCC in 2016. Our center therefore played an increasingly important role in facilitating the finding and evaluation of suspicious hepatic lesions and initiating individualized treatment plans. The relatively high rate of Child–Pugh A, BCLC-A, ALBI I, and significantly lower age at the time of diagnosis in the second time period might reflect the successful implementation of improved screening programs. 

Regarding the treatment of HCC, 2018 marked the year of significant changes in the medical management of HCC after the approval and widespread availability of new immunotherapies within the European Union (EU) [27]. The combination of Atezolizumab and Bevacizumab, newly introduced in 2018, replaced Sorafenib as the mainstay of systemic treatment for advanced HCC [13,28]. The overall survival for the Atezolizumab/Bevacizumab combination was 67.2% compared to the 54.6% for Sorafenib alone, as well as a prolonged progression free survival of 6.8 months for the Atezolizumab/Bevacizumab combination compared to 4.3 months for Sorafenib [28].

The fact that less patients were staged as BCLC-B and more with BCLC-C in the second time period caused a treatment shift away from interventional therapies and led to an increasing number of patients being treated with systemic therapy. In other words, the concept of treatment-stage migration in BCLC-B caused a higher number of systemic therapies. 

The retrospective analysis of the treatment modalities in our cohort also revealed a significant difference between noncirrhotic and cirrhotic patients, where systemic treatment was the most common first-line treatment of noncirrhotic patients, compared to interventional treatment in the cirrhotic group. The majority of noncirrhotic patients were staged as BCLC-C, which made them unsuitable candidates for interventional treatment. 

Despite the introduction of the aforementioned novel substances, there was no significant improvement in the median overall survival time between the two time periods. However, OS still improved when compared to data collected in Austria over the past three decades. In a previous study conducted by our working group, regarding the epidemiology of HCC in Austria, patients diagnosed between 1990 and 1999, had a mean survival time of 2.6 months, those diagnosed between 2000 and 2009 had a 5.6-month survival time, and those diagnosed with HCC between 2010 and 2018 had an OS of 9.3 months (*p* < 0.001) [9].

The ALBI score represents a biochemical score, including serum bilirubin and albumin levels suggesting overall hepatic function [29]. Our noncirrhotic patients were shown to have better liver function (higher rate of ALBI grade 1) at the time of diagnosis, despite being diagnosed at a later stage of disease according to the BCLC system. In another single-center cohort study, it was also shown that despite well-preserved liver function, the OSs of noncirrhotic HCC patients were not better compared to cirrhotic HCC patients due to late detection of disease [30].

Unfortunately, only 13 patients in our cohort received an OLT. This might be due to previous, long waiting lists for liver transplantations, causing patients to progress to more advanced stages of disease while awaiting surgery. Some patients also regrettably continued to consume excessive amounts of alcohol and were therefore excluded from the transplantation list.

This study was limited by the small study population and retrospective data collection. Our study was, however, strengthened by the inclusion of objective study parameters, such as laboratory values. Patient data were well documented and easily accessible in the electronic record keeping system. Klinikum Klagenfurt is a tertiary center and the third biggest hospital in Austria. It therefore has a large drainage area and receives referrals from all over the province of Carinthia. 

## 5. Conclusions

The rising incidence of HCC warrants the identification of changes occurring in the epidemiology and underlying etiology of HCC. 

More patients have been diagnosed without underlying liver cirrhosis in recent years. This group of patients was particularly interesting, as it demonstrated a different etiological pattern compared to HCC patients with underlying liver cirrhosis. NASH secondary to lifestyle diseases played a bigger role in the development of HCC in patients without cirrhosis compared to alcoholic liver disease in cirrhotic patients. 

Since no improvement in OS was found, one could ask whether there is a need for new screening strategies to prevent a surveillance gap in noncirrhotic patients and improving survival outcomes in the future. 

## Figures and Tables

**Figure 1 cancers-15-05215-f001:**
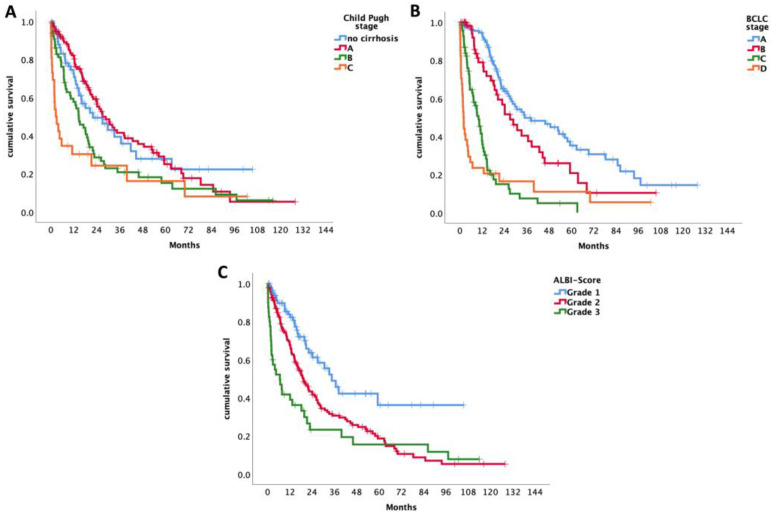
(**A**) Survival time of all patients including OLT according to Child–Pugh score. (**B**) Survival time of all patients including OLT according to BCLC. (**C**) Survival time of all patients including OLT according to ALBI score.

**Figure 2 cancers-15-05215-f002:**
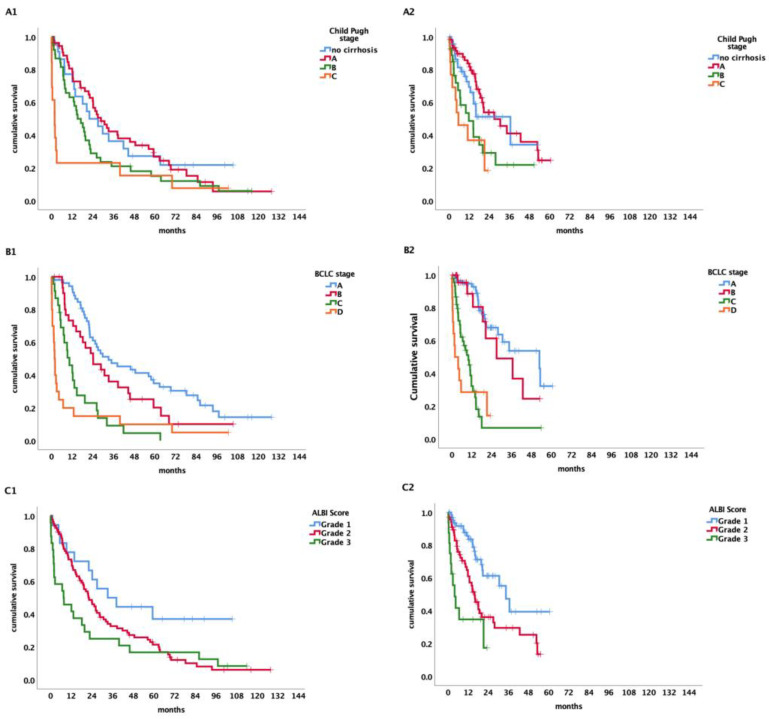
(**A**) Survival time of patients in time period 1 (**A1**) vs. time period 2 (**A2**) including OLT according to Child–Pugh score. (**B**) Survival time of patients in time period 1 (**B1**) vs. time period 2 (**B2**) including OLT according to BCLC. (**C**) Survival time of patients in time period 1 (**C1**) vs. time period 2 (**C2**) including OLT according to ALBI score.

## Data Availability

Data were obtained retrospectively from the Klinikum Klagenfurt am Wörthersees clinical information software system known as Orbis and the laboratory results were acquired from the electronic system, Lauris.

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
