# Peer review of "Changes in the Epidemiology of Hepatocellular Carcinoma in Carinthia, Austria, 2012–2023"

_cancers, 2023, doi:10.3390/cancers15215215_

Round 1
Reviewer 1 Report
Comments and Suggestions for Authors
Considering the comprehensive data published before (https://www.mdpi.com/2072-6694/14/13/3093), this local information should be published in another journal. However, I have some major comments to improve it:
ABSTRACT
1. Please state the case events in the results.
INTRODUCTION
1. The introduction could benefit from improved organization and flow. Consider dividing it into subsections to clearly introduce the significance of HCC, its global and Austrian context, risk factors, staging systems, and the specific objectives of the study. This would make it easier for readers to follow the logical progression of the information.
2. It would be beneficial to clearly state the specific research gap or problem that this study aims to address. Additionally, the study's objectives should be explicitly mentioned, so readers have a clear understanding of what the research intends to achieve.
3. It is important to emphasize why this particular study on HCC in Carinthia is significant. Discussing how the results could contribute to local healthcare improvements, prevention strategies, or patient care would help readers understand the broader implications of the research.
METHODS
1. Please explain more about the data source. Could you provide a reference for it?
2. The data collection should be explained in more detail.
DISCUSSION
1. What were your strengths and limitations?
Comments on the Quality of English LanguageMinor revision is needed.
Author Response
- Please state the case events in the results.
Thank you for the comment above. The abstract has been reviewed and edited accordingly. More statistical information was added to the results section.
INTRODUCTION
1. The introduction could benefit from improved organization and flow. Consider dividing it into subsections to clearly introduce the significance of HCC, its global and Austrian context, risk factors, staging systems, and the specific objectives of the study. This would make it easier for readers to follow the logical progression of the information.
Thank you for the detailed feedback on how the Introduction could be improved. We have followed the above given advise and divided the introduction into subsections in order give the Introduction more structure and to ensure a logical flow of information.
- It would be beneficial to clearly state the specific research gap or problem that this study aims to address. Additionally, the study's objectives should be explicitly mentioned, so readers have a clear understanding of what the research intends to achieve.
We have expanded our study objectives subsection in order to explain our aims in more detail. We have now stated that there is a lack of data on the incidence NASH in our region. We have explained that this lack of data might lead to a surveillance gap in the timeous detection of HCC as previous screening programs were aimed specifically at cirrhotic patients with underlying alcoholic liver disease. We have mentioned that HCC patients still are diagnosed at a late stage of disease have a poor prognosis and that more should be done to diagnose patients early. This can however only be done when we are aware of current epidemiological data in our region.
- It is important to emphasize why this particular study on HCC in Carinthia is significant. Discussing how the results could contribute to local healthcare improvements, prevention strategies, or patient care would help readers understand the broader implications of the research.
Thank you for the constructive feedback. We have added that there is a lack of epidemiological data in Carinthia. Even though Carinthia is situated in Austria, which is a first world country, data might differ as it is more rural. It is therefore necessary to know what changes are happening in our specific region in order to adjust prevention strategies accordingly. Seeing that the incidence of HCC is rising and patients generally having a poor outcome, it is important to know how we can adjust current screening programs to detect patients at an earlier stage of disease.
METHODS
1. Please explain more about the data source. Could you provide a reference for it?
Unfortunately, we cannot provide a reference for the data source. We used the hospital’s own electronic software systems known as Orbis. All clinical information of patients is loaded onto the system. Notes are all kept electronically instead of paper files. We looked up each patient’s electronic “file” individually to find the specific data points that we were looking for and captured this data on an Excel database.
- The data collection should be explained in more detail.
Thank you for this. We have elaborated on the data collection in the method section as explained above.
DISCUSSION
1. What were your strengths and limitations?
Thank you for alerting us that the strengths and limitations were missing from our manuscript. We have added this section to the discussion and pointed out that although our study was limited by a small population sample and retrospective character, we still have a large drainage area and stated that Klinikum Klagenfurt acts as a tertiary centre and is the third biggest hospital in Austria. We also added objective parameters to our data collection despite the study being conducted in a retrospective manner, which strengthens the validity of our findings.
Comments on the Quality of English Language
Minor revision is needed.
Thank you for the remark. We have noticed some languages errors and corrected them to the best of our abilities throughout the paper.
Reviewer 2 Report
Comments and Suggestions for Authors
Line 26 aetiology, It is not seen in the article.
Result from Klinikum Klagenfurt am Wörthersee Not generalizable to Australia (in title).
Line 28 observational not needed. Why Two time-periods??
Please add statistical method and sample size.
This: doi: 10.17179/excli2019-1842 can be useful for improving the introduction.
Line 100 2.1 Data retrieval is this type of study??
Line 104 Reference is needed.
Lin 106-110, why Two time-periods, While Tables 2 and 3, all patients are included in the analysis.
The method is not clear, it should be revised based on STROBE checklist: cohort studies.
a total of 285 patients were diagnosed with 136 HCC. 276 patients met the requirements of the inclusion, it’s ambiguous. Why excluded?
Is it important to compare two times?
The details of the analysis, the analysis method and its assumption should be stated.
In discussion, all sentences need reference. The current format is not acceptable.
Conclusions is long and in conclusion, reference is not needed.
Author Response
Line 26 aetiology, It is not seen in the article.
Thank you for the remark. We have specifically looked at the underlying etiology of HCC within our study population. In other first-world countries it has been said that the incidence of NAFLD/ NASH as the cause for HCC is rising. Traditionally in Austria, underlying alcoholic liver disease has always been the main etiological factor for liver cirrhosis and subsequently for HCC. It is important for us to know whether or not there is an upwards trend in the number of NASH cases as the data for our area on this topic is extremely limited. If there is a change in the underlying etiology within our population, it is important to know this in order to shift our screening programs in this direction and avoid a surveillance gap. Even though alcoholic liver disease remained the most common underlying cause of HCC, a clear upward trend in the number of NASH cases was evident as described in our results.
I our study aims section within the introduction, we have now added a paragraph to explain the relevance in more detail.
Result from Klinikum Klagenfurt am Wörthersee Not generalizable to Australia (in title).
Carintha is a province in Austria (Central Europe, not Australia) where Klinikum Klagenfurt is situated in the capital city of the province. It is a tertiary hospital and serves a large drainage area. Even though Austria is a first-world country, Carinthia still has a large rural population. It is therefore important for us to have an updated knowledge of the epidemiology of our region as it might differ to a more urban area such as Vienna where people have a more sedentary lifestyle in general.
Line 28 observational not needed. Why Two time-periods??
Thank for the feedback. We have removed the word “observational” from the manuscript. The two time-periods were created so that we can compare and comment on the changes that have happened over the past 12 years. By creating a group for comparison, we were able to observe specific changes in etiology during recent years. The two time-periods also allowed us to see whether survival has improved after 2018. In 2018 the EU allowed various immunotherapies on the market for the systemic treatment of HCC which showed promising results in overall survival in initial studies. Since our centre receives referrals from all over the province, often in advanced stages of disease we saw a major increase in the number of patients receiving immunotherapy as first-line treatment after 2018. We were then able to see whether or not there was an improvement in survival time after the widespread availability of immunotherapy.
Please add statistical method and sample size.
Thank you for the feedback. We have added the sample size in numbers to the Abstract and it can also be found in the first paragraph of the Results section. The statistical methods can be found in the Methods section where we have mention that the SPSS program was used. We have however expanded on the section to explain the statistical analysis in more detail. The statistical method has also been added to the abstract.
This: doi: 10.17179/excli2019-1842 can be useful for improving the introduction.
Thank you for this useful article. We worked on improving our introduction by giving it more structure with subheadings. We added more information and changed the paragraphs around to improve the flow and make it more logical for readers to follow.
Line 100 2.1 Data retrieval is this type of study??
Under the heading of Methods, we meant to explain how data was collected and therefore added the subheading of “data retrieval”. We understand now that this might be ambiguous for readers and changed the name of the subheading to “single cohort study”.
Line 104 Reference is needed.
Unfortunately, we cannot provide a reference for the data source. We used the hospital’s own electronic software systems known as Orbis. All clinical information of patients is loaded onto the system. Notes are all kept electronically instead of paper files. We looked up each patient’s electronic “file” individually to find the specific data points that we were looking for and captured this data on an Excel spreadsheet. We also cannot provide a reference for the hospital’s electronic laboratory systems as it is merely an internal electronic platform where patient laboratory results are kept.
Lin 106-110, why Two time-periods, While Tables 2 and 3, all patients are included in the analysis.
Thank you for the above mentioned remark. As explained in previous comment, the two time periods were created in order to showcase the presence or absence of a change in etiology of HCC in our setting. Other first world-countries have proven that NASH related HCC cases are rising, and we have data to prove that this is the case in Vienna. We however do not know what the trends of the underlying etiology of HCC in our more rural setting are. Without a comparison, we cannot prove that a change has occurred as very little data is available on the incidence of NASH in our area.
Thank you for pointing out that there is a problem with Tables 2 and 3.
As for table 2: We first calculated the overall survival time to have a general overview of what the survival outcome was for our cohort as a whole. In the text of the survival results itself, the difference between the time periods have also been commented on.
We have now recalculated the statistics for the survival time and made a new table to showcase the differences in survival time between the two time-periods. The results section with regards to survival has also been updated accordingly. We hope that by adding a supplementary table on the changes in survival over time, we will provide readers with a clearer understanding of its importance.
Table 3: Table three differs from table 1 and 2 as here we look at the difference between cirrhotic and non-cirrhotic HCC patients. After we found that the number of HCC patients without liver cirrhosis, increased significantly in the second time-period, we were interested in why this was the case. We wanted to look at this group of patients specifically to see whether there are any major differences between the two groups. Having a more in-depth knowledge of this group could contribute in guiding our prevention strategies for the future. It was clear that the incidence of NASH was much higher in the non-cirrhotic group, and this is very important information for planning future surveillance programs.
The method is not clear, it should be revised based on STROBE checklist: cohort studies.
Thank you, this feedback. We have edited our methods sections and mentioned more clearly how the data was collected. We have reviewed the STROBE checklist to ensure that all aspects are mentioned in the methods. We have also added our study strengths and limitations to the discussion section. There the concern of potential bias is addressed seeing that the study was conducted retrospectively. We included objective variables which helps to limited bias.
a total of 285 patients were diagnosed with 136 HCC. 276 patients met the requirements of the inclusion, it’s ambiguous. Why excluded?
A total of 285 patients were seen at Klinikum Klagenfurt in the given time-period. As mentioned in our exclusion criteria section under Methods, patients with mixed HCC and biliary tract tumors and missing data vital to our study aim was excluded from the criteria. If a patient for example had important laboratory values missing like Albumin or Bilirubin, they were excluded from the cohort. Without this data there were many variables that could not be calculated.
Is it important to compare two times?
As mentioned previously, we have limited data available on the current epidemiology and etiology in our setting. We therefore wanted to create two time-periods in order to comment on whether or not change has occurred in the underling etiology of NASH for example.
The details of the analysis, the analysis method and its assumption should be stated.
The statistical analysis was explained in the materials and methods section. We have added that the Cox regression was used first and thereafter the Kaplan-Meier methods for the graphs.
The general assumption was that it is believe that the incidence of NASH is rising in first-world countries and that we also want to know if this is true for our region.
In discussion, all sentences need reference. The current format is not acceptable.
Thank you for this feedback. In our discussion, we often comment on our own finding in our study and our interpretation thereof. When we compared it to other studies or data, we did reference it accordingly.
We have however added more referencing to our discussion.
Conclusions is long and in conclusion, reference is not needed.
The conclusion has been summarized to shorten it and make it more to the point. Refence has been removed.
Reviewer 3 Report
Comments and Suggestions for Authors
The phrase "Carinthia, Austria" is used in the title of the article. As an academic, I do not find it appropriate to write specific regions. If the region you are talking about is an important region for HCC, you can write that this region does not have a feature in this sense. If you have written all the epidemiological features of Austria about HCC, that is, if all Austrian data are used in the article, I can understand that too. However, looking at the text of the article, it is seen that such a situation does not exist.
The last sentence of the introduction is too long. This sentence should only explain the purpose of this study.
The epidemiological features of HCC should be emphasized in the introduction part of the article. The epidemiology of HCC in Austria should be included in the second-to-last paragraph of the introduction.
Epidemiological features should be written in more detail at the beginning of the article. In this regard, you can benefit from the following studies:
Kucukakcali Z, Akbulut S, Colak C. Machine Learning-based Prediction of HBV-related Hepatocellular Carcinoma and Detection of Key Candidate Biomarkers. Civil Med J. 2022;37(3):255-263.
Ince V, Sahin TT, Akbulut S, Yilmaz S. Liver transplantation for hepatocellular carcinoma: Historical evolution of transplantation criteria. WorldJ Clin Cases. 2022;10(29):10413-10427
In the methodology section, the demographic and clinical variables that will be examined in this study and the units of laboratory variables (such as U/L, mg/dl) should be given clearly.
As a transplant surgeon, I am disappointed that only 13 of the 285 HCC patients included in this study received a liver transplant. This situation must be explained in the text of the article.
A few sentences should be written about the limitations of this study.
If ethics committee approval has been obtained for this study, number and date information should be given.
Please write the word "etiology" instead of "Aetiology"
Why didn't you do Cox regression analysis for survival analysis?
Comments on the Quality of English LanguageI cannot comment clearly because my native language is not English.
Author Response
Quality of English Language: (x) Minor editing of English language required
We have reviewed our manuscript and corrected languages errors to the best of our knowledge.
The phrase "Carinthia, Austria" is used in the title of the article. As an academic, I do not find it appropriate to write specific regions. If the region you are talking about is an important region for HCC, you can write that this region does not have a feature in this sense. If you have written all the epidemiological features of Austria about HCC, that is, if all Austrian data are used in the article, I can understand that too. However, looking at the text of the article, it is seen that such a situation does not exist.
Thank you for the above mentioned remark. Carinthia is a province within Austria with a more rural population compared to a more urban area such as Vienna. Klinikum Klagenfurt is situated in the capital city of the province of Carinthia. It serves as a tertiary hospital and has a large drainage area from the countryside. The availability of data for our specific region with regards to epidemiology, etiology and survival outcome is extremely limited. It is for example known that the incidence of NASH if rising in many first-world countries and data is also available for Vienna where people tend to have a more sedentary lifestyle. We however do not know if this is also the case for our rural region in Carinthia. If there is a change the underlying etiology of HCC, it is important for policymakers in our specific region to be aware of this fact in order to change screening programs and diagnose patients at an early stage of disease, especially those who do not present with classical liver cirrhosis due to underlying alcoholic liver disease as has been the case in Austria for many years. Therefore, it is important to us to state that this study is applicable to our specific region.
The last sentence of the introduction is too long. This sentence should only explain the purpose of this study.
Thank you for this remark. We have revised our Introduction by adding subheadings to provide better structure and improved flow for readers to follow. The last paragraph has been edited and now explains the study objectives in a clearer, more detailed manner.
The epidemiological features of HCC should be emphasized in the introduction part of the article. The epidemiology of HCC in Austria should be included in the second-to-last paragraph of the introduction.
Thank you for the commentary. We have improved the structure of the Introduction. The epidemiology of HCC in Austria moved to downwards in the introduction, to allow for a more logical flow of information.
Epidemiological features should be written in more detail at the beginning of the article. In this regard, you can benefit from the following studies:
Kucukakcali Z, Akbulut S, Colak C. Machine Learning-based Prediction of HBV-related Hepatocellular Carcinoma and Detection of Key Candidate Biomarkers. Civil Med J. 2022;37(3):255-263.
Ince V, Sahin TT, Akbulut S, Yilmaz S. Liver transplantation for hepatocellular carcinoma: Historical evolution of transplantation criteria. WorldJ Clin Cases. 2022;10(29):10413-10427
Thank you for sharing the helpful studies. We have read both studies and used it to help as improve the epidemiology and background section of our manuscript.
In the methodology section, the demographic and clinical variables that will be examined in this study and the units of laboratory variables (such as U/L, mg/dl) should be given clearly.
Thank you for this remark. We have added the requested information to the methodology section.
As a transplant surgeon, I am disappointed that only 13 of the 285 HCC patients included in this study received a liver transplant. This situation must be explained in the text of the article.
Klinikum Klagenfurt is unfortunately not a transplant center, and we cannot perform any liver transplants at our facility. We however still act as a tertiary hospital for the rural province of Carinthia and have a large referral/ drainage area that we serve. It might therefore be that we receive more patients with advanced stages of HCC at our hospital.
Previously the waiting list for a liver transplant was very long and many of our patients progressed to a more advanced HCC stage during this waiting time.
In Austria underlying alcoholic liver disease is the most common cause of HCC. We also had patients on our transplant list who unfortunately continued to use excessive amounts of alcohol and therefore did not qualify for a transplant anymore.
We have added a paragraph in the discussion section the explain this occurrence.
A few sentences should be written about the limitations of this study.
Thank you for this valuable feedback. We have added a section at the end of our discussion that now describes the limitations and strengths of our study. Our study was limited by the small patient population and retrospective manner of collecting data.
If ethics committee approval has been obtained for this study, number and date information should be given.
Thank you for pointing this out. Our study did receive ethical approval by the Ethics Committee of Carinthia. The relevant information has been added.
Please write the word "etiology" instead of "Aetiology"
We have changed the word aetiology from its British spelling to the American spelling, etiology throughout the whole manuscript.
Why didn't you do Cox regression analysis for survival analysis?
Cox regression was used for survival analysis, but we failed to mention it. Thank you for highlighting this issue. We have added it to the statistical methods in our manuscript.
Round 2
Reviewer 1 Report
Comments and Suggestions for Authors
Although the authors have made revisions, the article does not yet meet the qualifications for publication in this journal.
Author Response
Thank you for your review.
We intensively revised the discussion section of our manuscript.
Reviewer 2 Report
Comments and Suggestions for Authors
Previous comments have not been modified.
Check previous comments.
Also, the study is local, and its results cannot be generalized.
Author Response

(The authors gave the same response as above.)

Reviewer 3 Report
Comments and Suggestions for Authors
I do not think it is appropriate to include a specific region (i.e. not a place known worldwide) in the article title. I leave this decision to the editor.
Author Response
Thank you for your review